# Dehydration of Bioethanol to Ethylene over H-ZSM-5 Catalysts: A Scale-Up Study

**Sanggil Moon [1], Ho-Jeong Chae [1,2],* and Min Bum Park [3],***

[1]  CO2 Energy Vector Research Group, Carbon Resources Institute, Korea Research Institute of Chemical Technology, Daejeon 34114, Korea; pulngil@krict.re.kr

[2]  Department of Green Chemistry and Biotechnology, University of Science and Technology, Daejeon 34113, Korea

[3]  Innovation Center for Chemical Engineering, Department of Energy and Chemical Engineering, Incheon National University, Incheon 22012, Korea

*   Correspondence: hjchae@krict.re.kr (H.-J.C.); mbpark@inu.ac.kr (M.B.P.); Tel.: +82-42-860-7290 (H.-J.C.); +82-32-835-8671 (M.B.P.)

**Abstract:** Bioethanol dehydration was carried out in a bench scale reactor-loaded H-ZSM-5 molded catalyst, which increased by tens of times more than at lab scale (up to 60 and 24 times based on the amount of catalyst and ethanol flow rate, respectively). From the results of the lab scale reaction, we confirmed the optimum Si/Al ratio (14) of H-ZSM-5, reaction temperature (~250 °C), and weight hourly space velocity (WHSV) (<5 h$^{-1}$) indicating high ethanol conversion and ethylene selectivity. Five types of cylindrical shaped molded catalysts were prepared by changing the type and/or amount of organic solid binder, inorganic solid binder, inorganic liquid binder, and H-ZSM-5 basis catalyst. Among them, the catalyst exhibiting the highest compression strength and good ethanol dehydration performance was selected. The bench scale reaction with varying reaction temperature of 245–260 °C and 1.2–2.0 h$^{-1}$ WHSV according to reaction time showed that the conversion and ethylene selectivity were more than 90% after 400 h on stream. It was also confirmed that even after the successive catalyst regeneration and the reaction for another 400 h, both the ethanol conversion and ethylene selectivity were still maintained at about 90%.

**Keywords:** bench scale; bioethanol; ethylene; H-ZSM-5; molded catalyst

## 1. Introduction

Along with extreme global warming caused by the use of fossil energies, securing alternative energy resources due to the depletion of fossil fuels has become a global issue at present. As one of the alternatives, the production of biofuels by the catalytic conversion of biomass has attracted much attention [1,2]. For example, since the global aviation industries account for around 2% of total $CO_2$ emissions, they are making efforts to meet the greenhouse gas reduction obligations through the use of biomass-derived jet fuel [3,4]. Byogy Renewables Inc. has reported the production of jet fuel from bioethanol [5]. Ethanol can be converted to ethylene by dehydration over an acid catalyst like zeolite, silica alumina, alumina, etc. [6,7]. The dehydrated ethanol can be further turned into the long chain hydrocarbons by oligomerization [8].

Bioethanol is generally produced by the biomass fermentation process and contains a large amount of water. Dehydration by distillation can produce high-purity bioethanol, while it requires a lot of energy consumption. Thus, the direct use of bioethanol containing 10% or more water is one of the industrially important issues. The use of H-ZSM-5 zeolite (framework type MFI) catalyst replacing a conventional alumina catalyst enabled higher activity in ethanol dehydration even at

lower temperature, despite the use of hydrous ethanol [6,7]. In order to increase the catalytic stability, some post-treated H-ZSM-5 catalysts, such as dealumination by acid treatment [9], desilication by base treatment [9], phosphorus and/or lanthanum loading [10–12], transition metal ion exchange [13] etc., have been tested in ethanol dehydration. In addition, the use of a nano crystalline H-ZSM-5 or H-ZSM-5/H-SAPO-34 core-shell structure has been reported to improve molecular diffusion [14,15].

However, most of these previous studies were performed at lab scale, and a few studies on ethylene production by ethanol dehydration over a bench scale have been reported in the literature [16]. In this study, we explored the bioethanol dehydration for the production of ethylene over H-ZSM-5 acid catalysts at both lab and bench scales. We prepared five types of molded catalysts by using H-ZSM-5 as a basis catalyst and compared the bioethanol dehydration performance with especial emphasis on the activity, stability, and regenerability of catalyst.

## 2. Results and Discussion

### 2.1. Lab Scale Bioethanol Dehydration

#### 2.1.1. Effect of Si/Al Ratio of H-ZSM-5, Reaction Temperature, and Weight Hourly Space Velocity (WHSV) in Ethanol Dehydration

Figure 1 shows the ethanol conversion and ethylene selectivity obtained after ethanol dehydration for 20 h at 250 °C and 5 h$^{-1}$ WHSV over H-ZSM-5 catalysts with different Si/Al ratios. The Si/Al ratio of H-ZSM-5 after calcination became somewhat higher than that of the parent NH$_4$ form (see below), but here it was denoted with the SiO$_2$/Al$_2$O$_3$ value of the NH$_4$ form provided by the vendor. All the H-ZSM-5 catalysts prepared here almost maintained their initial conversion and ethylene selectivity during the reaction time of 20 h. They exhibited similar levels of conversion and ethylene selectivity except those from the catalyst with SiO$_2$/Al$_2$O$_3$ = 280. The lower the Si/Al ratio, the higher the conversion and ethylene selectivity. It is well known that the ethanol dehydration is a typical Brønsted acid catalyzed reaction [7,17,18]. Thus, the increased reactivity should be due to the increase of acid site density according to the increase of Al content in zeolite framework. In this study, we carried out the rest of the study using ZSM-5 with SiO$_2$/Al$_2$O$_3$ = 23 and its acid catalyst was named as H-ZSM-5A.

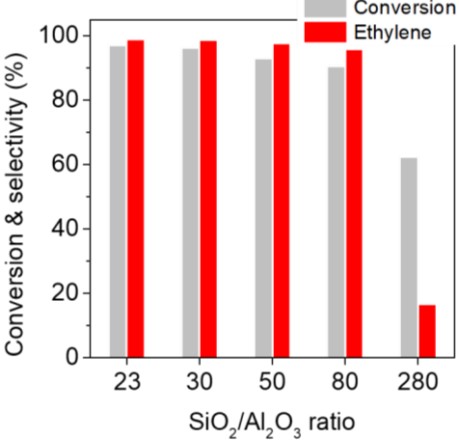

**Figure 1.** Ethanol conversion (■) and selectivity to ethylene (■) obtained after 20 h on stream as a function of SiO$_2$/Al$_2$O$_3$ ratio of H-ZSM-5 in dehydration of bioethanol at 250 °C and 5 h$^{-1}$ WHSV (0.2 g of catalyst and 0.021 mL min$^{-1}$ of ethanol).

Figure 2 shows the conversion and product selectivities obtained after 20 h on stream over 0.2 g of H-ZSM-5A according to the reaction temperature (210–350 °C) and WHSV (5, 8, and 10 h$^{-1}$). Regardless of the WHSV, the conversion increased as the reaction temperature increased. In addition, it can be seen that the lower the WHSV, the higher the conversion at low temperature region, and

the lower the increase rate of conversion according to the temperature increase. At 8 and 10 h$^{-1}$ WHSV, the selectivity to C$_4$+ was almost zero, while the selectivities to ethylene and diethyl ether increased and decreased, respectively, with increasing reaction temperature. This may be because the intermediate product diethyl ether was obtained as a more stable product at lower temperature during the ethanol dehydration [19–21]. On the other hand, in the case of 5 h$^{-1}$ WHSV, although the ethylene selectivity was high even at a relatively lower temperature, the selectivities of ethylene and C$_4$+ were drastically decreased and increased, respectively, at a relatively higher temperature. It can be inferred that ethylene formed from the dehydration of ethanol was further converted to long-chain hydrocarbons at high temperature by oligomerization on zeolite acid sites [8]. Therefore, it can be concluded that WHSV lower than 5 h$^{-1}$ and about 250 °C of temperature are the optimal conditions to obtain high ethanol conversion and ethylene selectivity.

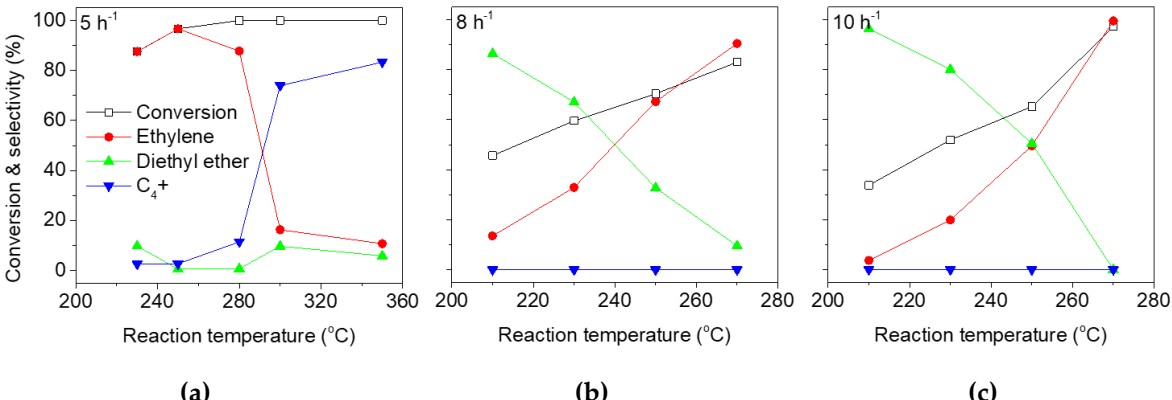

**Figure 2.** Ethanol conversion (□) and selectivities to ethylene (●), diethyl ether (▲), and C$_4$+ compounds (▼) obtained after 20 h on stream as a function of reaction temperature in dehydration of bioethanol over 0.2 g of H-ZSM-5 (SiO$_2$/Al$_2$O$_3$ = 23) at 5 (left), 8 (middle), and 10 h$^{-1}$ (right) WHSV.

### 2.1.2. Preparation of Modified H-ZSM-5 Catalysts and Their Dehydration of Ethanol

In order to investigate the dehydration of ethanol according to the modification of physicochemical properties of H-ZSM-5A, we prepared three more catalysts treated with sodium hydroxide (H-ZSM-5B), ammonium hexafluorosilicate (AHFS) (H-ZSM-5C), and pseudomorphic synthesis (H-ZSM-5D). It is well known that the three methods can form additional mesopores by desilication, dealumination, and recrystallization of zeolite structure, respectively [9,22–24].

Only the relative intensity of X-ray peaks for the three modified catalysts was slightly different from that of parent H-ZSM-5A (Figure S1). Thus, their structures were maintained without significant changes even after the treatments. As shown in Table 1, the Si/Al ratio (14) of H-ZSM-5A became slightly higher than that of the original value (11.5) in the course of acid catalyst formation by calcination of its NH$_4$ form. It can confirm that the Si/Al ratios of H-ZSM-5B and H-ZSM-5C were slightly lower (12) and higher (15) by desilication and dealumination, respectively, as expected. On the other hand, H-ZSM-5D had a lower Si/Al ratio (10) even than that of NH$_4$-ZSM-5, which can be understood that the Al content of zeolite framework should be increased in order to balance the charge of organic cations (tetramethylammonium (TMA$^+$) and/or cetyltrimethylammonium (CTA$^+$)) additionally used in the psedomorphic recrystallization [24]. After the three treatments, the total BET surface areas of the samples were somewhat lower than that of the parent catalyst, but the external surface area and pore volume were relatively increased, indicating that a part of the mesopores was formed.

**Table 1.** Physicochemical properties of parent and modified H-ZSM-5 catalysts.

| Catalyst | Si/Al [1] | BET Area (m$^2$ g$^{-1}$) [2] | | | Pore Volume (cm$^3$ g$^{-1}$) [4] | Relative Acid Site Density [5] | | | |
|---|---|---|---|---|---|---|---|---|---|
| | | Total | External [3] | Microporous | | Total | Weak | Medium | Strong |
| H-ZSM-5A | 14 | 425 | 104 | 321 | - | 100 | 51 (186) | 2 (283) | 47 (383) |
| H-ZSM-5B | 12 | 384 | 141 | 243 | 0.36 | 93 | 39 (171) | 54 (253) | 0 (-) |
| H-ZSM-5C | 15 | 394 | 108 | 286 | 0.20 | 98 | 44 (187) | 5 (304) | 49 (391) |
| H-ZSM-5D | 10 | 374 | 114 | 260 | 0.27 | 92 | 42 (178) | 9 (242) | 41 (366) |

[1] Determined by elemental analysis. [2] Calculated from $N_2$ sorption data. [3] Determined according to the *t*-plot method. [4] Calculated using the BJH formalism from the $N_2$ adsorption branch isotherm. [5] Determined from peak decomposition of $NH_3$ temperature programmed-desorption (TPD) data using PeakFit curve-fitting program. The values given in parentheses are the temperature maxima (°C) of each $NH_3$ desorption peak.

Figure 3 compares the $NH_3$ temperature programmed-desorption (TPD) profiles of the parent and three modified H-ZSM-5 catalysts. Three regions of weak, medium, and strong acid sites were roughly distinguished centered around 200, 300, and 400 °C, respectively. Despite the change of Si/Al ratios after the post-treatments, there were no significant differences in the amount of total acid sites (Table 1), because the Si/Al ratios of four catalysts are not much different within 10–15. However, the change of acid strength was clearly observed by the modification of relative Si or Al content in H-ZSM-5 zeolite framework. Due to the relatively increased Al content of H-ZSM-5B and H-ZSM-5D, the amount of strong acid sites was reduced compared to the parent H-ZSM-5A. In particular, in the case of H-ZSM-5B, the strong acid sites almost disappeared due to desilication and the distribution of medium acid sites near 300 °C was newly generated. Furthermore, it can be seen that the strong acid site was slightly increased by the relatively decreased Al content in H-ZSM-5C. These results are in good agreement with the distribution of acid sites according to Si/Al ratio of general zeolite catalysts [25].

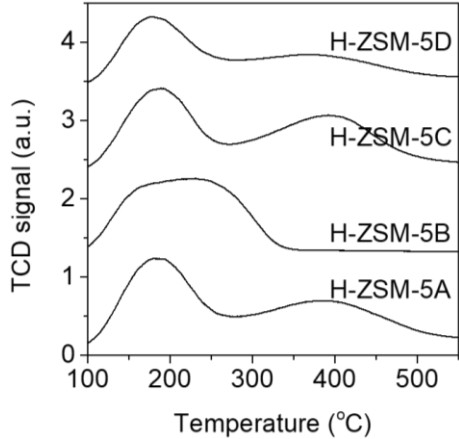

**Figure 3.** $NH_3$ TPD curves of H-ZSM-5A, H-ZSM-5B, H-ZSM-5C, and H-ZSM-5D.

Figure 4. presents the conversion and product selectivities obtained from the four catalysts (H-ZSM-5A–D) at 260 °C after 20 h on stream with changes in WHSVs (2, 5, and 10 h$^{-1}$). Like the above results in Figure 2, both higher ethanol conversion and ethylene selectivity were observed for all the catalysts at lower WHSV. At 2 h$^{-1}$ WHSV, all the catalysts exhibited conversions close to 100% and ethylene selectivities to 80% or more, while the reactivity deteriorated at 5 and 10 h$^{-1}$ as the reaction conditions became severe. At 5 and 10 h$^{-1}$ WHSV, H-ZSM-5B composed with weak and medium acid sites rather than strong by desilication showed a lower ethylene selectivity than H-ZSM-5A, and the selectivity to diethyl ether increased by more than two times. It has been reported that diethyl ether is more selective than ethylene on weaker zeolite acid sites in ethanol dehydration [19–21]. We should note here that the parent H-ZSM-5A with no change in acidity and mesoporosity exhibited the best performance with an ethanol conversion of more than 80% and an ethylene selectivity higher than

70% even at 10 h$^{-1}$ WHSV. Figure 5 compares the conversions and ethylene selectivities obtained from the four catalysts when the reaction temperature was increased from 245 to 265 °C by 5 °C at 4 h intervals according to the time on stream (TOS) under the constant 2 h$^{-1}$ WHSV. Unlike the other modified catalysts, the parent H-ZSM-5A also showed the best activity, i.e., nearly 100% conversion and steady selectivity to ethylene more than 90% during the total reaction time of 20 h. Therefore, the change of framework Si/Al ratio and mesopore formation of H-ZSM-5 catalysts by the post-treatments performed in this study are not helpful for high ethylene yield in ethanol dehydration. As seen in the lower BET surface areas (Table 1), because of the instability of the structure after the post-treatments, the reactivity according to TOS were less favorable than that of parent H-ZSM-5A.

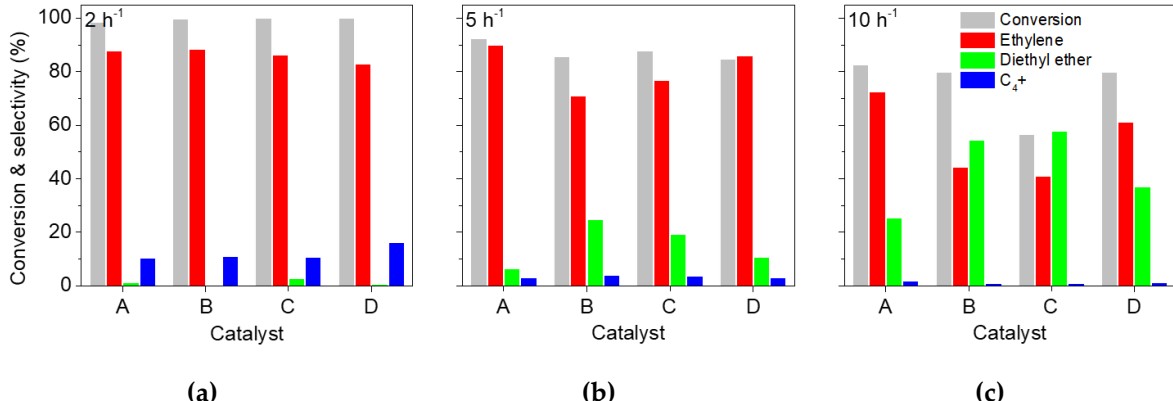

**(a)** **(b)** **(c)**

**Figure 4.** Ethanol conversion (▨) and selectivities to ethylene (■), diethyl ether (■), and C$_4$+ compounds (■) obtained after 20 h on stream in dehydration of bioethanol over 0.5 g of H-ZSM-5A, H-ZSM-5B, H-ZSM-5C, and H-ZSM-5D at 260 °C and 2 (left), 5 (middle), and 10 h$^{-1}$ (right) WHSV.

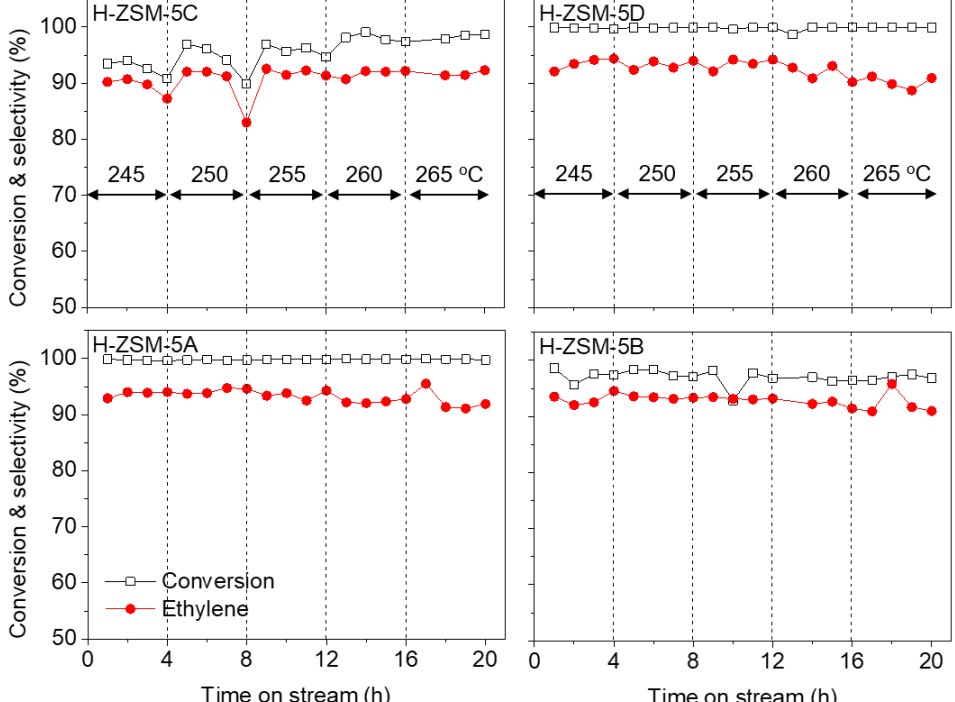

**Figure 5.** Ethanol conversion (□) and selectivity to ethylene (●) as a function of time on stream (TOS) in dehydration of bioethanol over 0.5 g of H-ZSM-5A, H-ZSM-5B, H-ZSM-5C, and H-ZSM-5D at–265 °C and 2 h$^{-1}$ WHSV.

## 2.2. Bench Scale Bioethanol Dehydration

### 2.2.1. Preparation of Molded H-ZSM-5A Catalysts and their Dehydration of Ethanol

Based on the results of lab scale studies, we have synthesized molded catalysts adaptable to bench and pilot scales by using H-ZSM-5A as a basis catalyst and compared their reactivity in ethanol dehydration. Five types of cylindrical shaped molded catalysts (H-ZSM-5A1–A5) with diameters of 3 mm and height of 5 mm were prepared following the typical synthesis method of a molded catalyst with a few variables [26], i.e., in an identical amount of organic solid binder (MC-40H), varying the amount of H-ZSM-5A basis catalyst, three types of inorganic solid binder (kaolinite, montmorillonite, or kaofine), and two types of inorganic liquid binder (silica or alumina sol) and their amount (Table 2). As shown in Table 2, the BET surface areas of all the molded catalysts were reduced compared to the basis H-ZSM-5A catalyst, and especially the proportion of external surface area was significantly reduced. As a result of measuring the compression strength of the molded catalysts, which is one of the most important factors in the scale-up catalytic reaction process, the strength was higher when silica sol was used rather than alumina sol (H-ZSM-5A1 vs A2), and it was also increased when increasing the proportion of silica sol (H-ZSM-5A2 vs A3). Depending on the type of inorganic solid binders, the strength was the highest when using montmorillonite (H-ZSM-5A3 vs A4 vs A5). Therefore, H-ZSM-5A4 composed with H-ZSM-5A:montmorillonite:silica sol:MC-40H = 60:10: 30:4 g showed the highest compression strength of 30 and 24 N among the molded catalysts prepared in this study. These are similar values than those of the molded catalysts available in general catalytic reaction process [26].

**Table 2.** Synthesis and characterization of molded catalyst.

| Molded Catalyst [1] | Synthesis Composition (g) [2] | | | | Physicochemical Properties | | | | | |
| | H-ZSM-5A | Inorganic Solid | Inorganic Liquid | Organic Solid (MC-40H) | BET Area ($m^2\ g^{-1}$) [3] | | | Pore Volume ($cm^3\ g^{-1}$) [5] | Compression Strength (N) [6] | |
| | | | | | Total | External [4] | Microporous | | | |
|---|---|---|---|---|---|---|---|---|---|---|
| H-ZSM-5A1 | 80 | Kaolinite, 10 | Ludox AS-40, 10 | 4 | 337 | 48 | 289 | 0.22 | 9 | 9 |
| H-ZSM-5A2 | 80 | Kaolinite, 10 | AS-520, 10 | 4 | 342 | 43 | 298 | 0.23 | 4 | 3 |
| H-ZSM-5A3 | 60 | Kaolinite, 10 | Ludox AS-40, 30 | 4 | 302 | 48 | 254 | 0.25 | 15 | 10 |
| H-ZSM-5A4 | 60 | Montmorillonite, 10 | Ludox AS-40, 30 | 4 | 327 | 85 | 241 | 0.29 | 30 | 24 |
| H-ZSM-5A5 | 60 | Kaofine, 10 | Ludox AS-40, 30 | 4 | 307 | 57 | 249 | 0.26 | 23 | 14 |

[1] Cylindrical shape with 3 mm in diameter and 5 mm in height. [2] 100–200 mL of deionized water was mixed together in each preparation. [3] Calculated from $N_2$ sorption data. [4] Determined according to the *t*-plot method. [5] Calculated using the BJH formalism from the $N_2$ adsorption branch isotherm. [6] Determined by universal testing machine (UTM).

Although the catalytic activity of molded catalysts prepared here is not a big difference, H-ZSM-5A2 showed the best performance (Figure S2). However, as shown in Table 2, since H-ZSM-5A2 showed the lowest compression strength (4 and 3 N), H-ZSM-5A4, which had the highest compression strength and exhibited the second best catalytic activity, is the most suitable molded catalyst for the scale-up catalytic reaction.

### 2.2.2. Long-Term Stability and Regeneration Tests of Molded H-ZSM-5A4 Catalyst

Figure 6 shows the conversion and product selectivities according to TOS until 300 h at constant reaction conditions of 260 °C and 2 h$^{-1}$ WHSV by loading 12 g of H-ZSM-5A4 molded catalyst into a bench scale reactor set. The conversion gradually decreased from almost 100% and dropped to less than 90% after 200 h on stream. Before 10 h on stream, the selectivities to ethylene and C$_4$+ rapidly increased and decreased, respectively, which may be due to the further oligomerization of ethylene formed by the ethanol dehydration at the early stage of reaction [8]. After that, the ethylene selectivity gradually decreased from about 95%, and after 250 h on stream, it decreased to 80% or less and the decreasing rate became rapid. On the other hand, the selectivity to diethyl ether tends to increase sharply after 250 h on stream, which may be come from the weakened strength of acid sites during the deactivation [19,20].

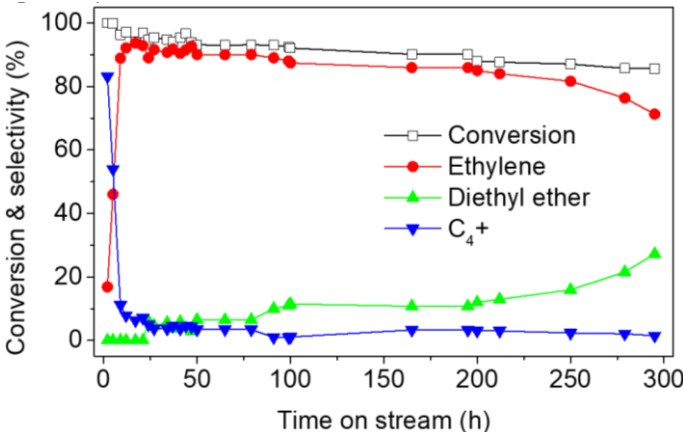

**Figure 6.** Ethanol conversion (□) and selectivities to ethylene (●), diethyl ether (▲), and C$_4$+ compounds (▼) as a function of TOS in dehydration of bioethanol over 12 g of molded H-ZSM-5A4 catalyst at 260 °C and 2 h$^{-1}$ WHSV.

It has been known that lanthanum containing zeolite acid catalyst has the effect of increasing amount of acid sites and maintaining high stability [11,12]. Thus, in order to maintain a constant conversion and ethylene selectivity with >90% during >300 h, 0.3 wt % La/H-ZSM-5A4 molded catalyst was prepared, and as shown in Figure 7, the conversion and product selectivities were examined by changing the reaction temperature (245–260 °C) and WHSV (1.2–2 h$^{-1}$) according to TOS. First, in order to improve the low ethylene and high C$_4$+ selectivities before 10 h on stream at 260 °C and 2 h$^{-1}$ WHSV as observed in Figure 6, the reaction temperature was lowered to 245 °C for the first 2 h (the bottom of Figure 7). As expected, the selectivity to C$_4$+ decreased sharply (~0%), but low conversion with around 70% and relatively high selectivity to diethyl ether (~20%) were observed due to the low temperature [19,20]. As the reaction temperature was gradually increased, the conversion and selectivity to ethylene were also gradually increased, and both recovered more than 90% after 7 h on stream at 260 °C and 2 h$^{-1}$ WHSV. When maintaining the reaction under the identical conditions given above, the conversion and selectivity to ethylene were gradually decreased like those observed in Figure 6. After 90 h on stream, the WHSV was gradually reduced to 1.8, 1.5, and 1.2 h$^{-1}$, and thus a high conversion and selectivity to ethylene maintained over 90% up to over 400 h on stream. However,

even at the mild reaction condition of 1.2 h$^{-1}$ WHSV, conversion and selectivity to ethylene tended to decrease sharply after 400 h on stream.

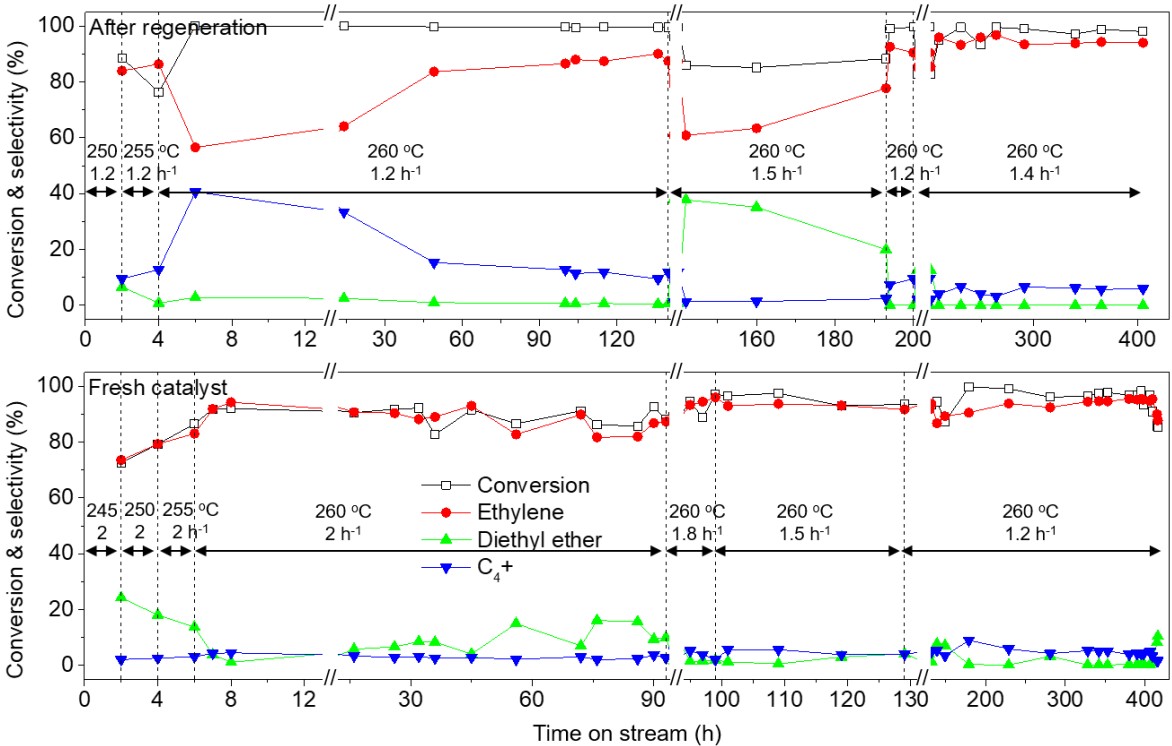

**Figure 7.** Ethanol conversion (□) and selectivities to ethylene (●), diethyl ether (▲), and C$_4$+ compounds (▼) as a function of TOS in dehydration of bioethanol over 12 g of 0.3 wt. % of lanthanum loaded La/H-ZSM-5A4 catalyst at 245–260 °C and 1.2–2 h$^{-1}$ WHSV (bottom), and successive reaction under identical conditions as given above after regeneration of catalyst in air flow (100 mL min$^{-1}$) at 550 °C (ramping rate of 10 °C min$^{-1}$) for 6 h right after previous sequence (top).

After a total of ca. 420 h on stream, the catalyst was regenerated by calcination at 550 °C for 6 h under flowing air and the dehydration reaction was successively carried out under similar conditions. As shown in the top of Figure 7, similar to the results obtained above, the conversion and product selectivities can be controlled by chaining the reaction temperature and WHSV. As a result, it was confirmed that after the regeneration of catalyst, the conversion and selectivity to ethylene were still maintained over 90% at 260 °C and 1.4 h$^{-1}$ WHSV even after 400 h on stream.

## 3. Materials and Methods

### 3.1. Catalyst Preparation

The NH$_4$ form of ZSM-5 zeolites with five different SiO$_2$/Al$_2$O$_3$ ratios (23, 30, 50, 80, and 280) were purchased from Zeolyst (CBV 2314, CBV 3024E, CBV 5524G, CBV 8014, and CBV 28014, respectively). The commercial ZSM-5 zeolites were converted to proton forms (H-ZSM-5) by calcination at 550 °C for 2 h.

Some portions of calcined CBV 2314 ZSM-5 (H-ZSM-5A) were refluxed in 1 M sodium hydroxide (Sigma-Aldrich, 50% in H$_2$O) aqueous solution (1 g solid per 10 mL solution) for 4 h at 70 °C (H-ZSM-5B), and separately AHFS (Sigma-Aldrich, 98%) aqueous solution (AHFS/Al = 1 mol/mol) for 3 h at 80 °C (H-ZSM-5C) under stirring to modify the physicochemical properties of H-ZSM-5 [9,22,23]. In addition, pseudomorphic H-ZSM-5D was prepared from another portion of H-ZSM-5A according to the procedure described in the literature [24]. The H-ZSM-5A was first treated with TMAOH (Sigma-Aldrich, 25% in H$_2$O) aqueous solution (OH$^-$/H-ZSM-5A = 0.003 mol g$^{-1}$) for 30 min at 35 °C,

and CTABr (Sigma-Aldrich, 99+%) was added to the mixture (CTABr/H-ZSM-5A = 1 g/g) and stirred for another 30 min at 35 °C. The final mixture was then hydrothermally treated at 150 °C for 20 h in the Teflon-lined autoclave. All the post-treated samples were calcined at 550 °C for 6 h before use as catalysts.

Five types of molded catalysts (H-ZSM-5A1–A5) were prepared from the combination of H-ZSM-5A as a parent catalyst, Kaolinite (Sigma-Aldrich), Montmorillonite (Sigma-Aldrich), or Kaofine (Thiele Kaolin) as an inorganic solid binder, Ludox AS-40 (Sigma-Aldrich, 40% in $H_2O$) or AS-520 (Nissan Chemicals) as an inorganic liquid binder, and MC-40H (Samsung Fine Chemicals) as an organic solid binder [26]. The details of each synthesis composition were summarized in Table 2. First, H-ZSM-5A, inorganic solid, and organic solid were uniformly mixed together. Then, inorganic liquid and deionized water were added into the solid mixture to form a paste. The paste mixture was injected into the molding machine (customized twin screw kneader), and the molded paste was dried at 110 °C overnight to evaporate the water. Finally, the dried pellet was calcined at 550 °C for 6 h before use as a catalyst. The final molded catalyst was a cylindrical shape with 3 mm in diameter and 5 mm in height. In order to prepare the lanthanum containing H-ZSM-5A4 molded catalyst, La/H-ZSM-5A was first prepared by incipient wetness impregnation with lanthanum acetate (Sigma-Aldrich, 99.9%) precursor solution to yield 0.3 wt. % of lanthanum. Then, the identical H-ZSM-5A4 molding process was performed as given above.

### 3.2. Catalyst Characterization

Powder X-ray diffraction (XRD) patterns were recorded on a Rigaku Multiplex diffractometer operated at 40 kV and 40 mA with Cu $K_\alpha$ X-ray radiation. Elemental analysis and $N_2$ sorption experiments of the catalysts studied here were carried out by a Thermo Scientific iCAP 6300 Duo inductively coupled plasma optical emission spectrometer and a Micromeritics Tristar II 3000 analyzer, respectively. $NH_3$ TPD was carried out on a fixed bed, flow-type apparatus linked to a thermal conductivity detector. A sample of ca. 0.05 g was activated in flowing He (50 mL min$^{-1}$) at 550 °C for 2 h. Then, 10 wt.% $NH_3$ was passed over the sample at 100 °C for 30 min. The treated sample was subsequently purged with He at the same temperature for 1 h to remove physisorbed $NH_3$. Finally, the TPD was performed in flowing He (30 mL min$^{-1}$) from 100 to 600 °C at a temperature ramp of 10 °C min$^{-1}$. The compression strength of molded catalysts was determined by UTM (Instron model 4482) [26].

### 3.3. Catalysis

As a reactant, purified bioethanol including ca. 10 wt.% of water was supplied from KAI. We used the bioethanol as supplied without any pretreatment. Lab scale ethanol dehydration was carried out in a homemade continuous-flow apparatus with a stainless steel fixed-bed micro reactor with 9.5 mm O.D., 7.0 mm I.D., and 200 mm length. Prior to the experiments, the catalyst was routinely activated under flowing $N_2$ (10 mL min$^{-1}$) at 550 °C for 2 h and then kept at desired reaction temperature (210–350 °C). The bioethanol was preheated at 200 °C and fed at a rate of 0.021–0.11 mL min$^{-1}$ using a high-performance liquid chromatography (HPLC) pump (Young-Lin SP930D) into the reactor containing 0.2 or 0.5 g of powder type catalyst (2–10 h$^{-1}$ WHSV) with $N_2$ 10 mL min$^{-1}$ as a carrier gas. Bench scale reaction was performed in another fixed-bed set with 12.7 mm O.D., 10.9 mm I.D., and 700 mm length of stainless steel reactor containing 2 or 12 g of molded catalyst with 0.21–0.51 mL min$^{-1}$ of bioethanol (1.2–5 h$^{-1}$ WHSV) at 245–260 °C. When we loaded the catalyst in the reactor, we tried to minimize the wall effect by tapping it. The temperature was measured at three points along the longitudinal axis of reactor and controlled within ±2 °C. After leaving the reactor, the products were analyzed on-line in a Young-Lin YL6500 gas chromatograph equipped with a HP-PLOT/Q capillary column (30 m x 0.535 mm x 40 μm) connected to flame ionization detector. Conversion was defined as the mole percentage of ethanol consumed during the reaction, and the selectivity of each product was calculated as the carbon mole percentage of each product over all the other products except ethanol.

## 4. Conclusions

In this study, we explored the bioethanol dehydration for the production of ethylene over H-ZSM-5 acid catalyst in both lab and bench scales. From the lab scale reaction, H-ZSM-5A with the lowest Si/Al ratio among the tested ZSM-5 ($SiO_2/Al_2O_3$ = 23–280) showed the highest ethanol conversion and ethylene selectivity under the identical conditions. In addition, the temperature around 250 °C and WHSV of less than 5 $h^{-1}$ exhibited the best catalytic activity. The physicochemical properties of H-ZSM-5A could be modified by three post-treatments (i.e., desilication, dealumination, and psedomorphic synthesis), while the catalytic activities of three modified catalysts were lower than that of parent H-ZSM-5A. For bench and pilot scale reactions, five types of molded catalysts were prepared by using H-ZSM-5A as a basis catalyst and controlling the type and/or amount of organic solid binder, inorganic solid binder, and inorganic liquid binder. It was found that H-ZSM-5A4 molded catalyst synthesized at a weight ratio of H-ZSM-5A: montmorillonite:silica sol:MC-40H = 60:10:30:4 showed the highest compression strength and good ethanol dehydration activity among the prepared molded catalysts. In order to further enhance the catalytic stability, La/H-ZSM-5A4 molded catalyst containing 0.3 wt.% of La was finally prepared. In the bench scale fixed-bed reactor-loaded 12 g of this catalyst, the long-term stability and the reaction after regeneration of catalyst were tested by varying reaction temperature (245–260 °C) and WHSV (1.2–2.0 $h^{-1}$) according to TOS. We confirmed that not only the catalytic activity can be controlled by changing the temperature and WHSV but also both the ethanol conversion and ethylene selectivity were higher than 90% during the reaction for about 400 h, and those two values still maintained 90% for another 400 h even after successive catalyst regeneration.

**Supplementary Materials:** The following are available online at http://www.mdpi.com/2073-4344/9/2/186/s1, Figure S1: Powder XRD patterns of H-ZSM-5A, H-ZSM-5B, H-ZSM-5C, and H-ZSM-5D, Figure S2: Ethanol conversion obtained after 3 h on steam in dehydration of bioethanol over 2 g of molded H-ZSM-5A1, H-ZSM-5A2, H-ZSM-5A3, H-ZSM-5A4, and H-ZSM-5A5 catalysts at 250 °C and 5 $h^{-1}$ WHSV.

**Author Contributions:** Conceptualization, H.-J.C.; Experiment, S.M.; Data analysis, S.M. and M.B.P.; Writing—original draft preparation, S.M. and M.B.P.; writing—review and editing, H.-J.C. and M.B.P.; Supervision, H.-J.C. and M.B.P.; Project administration, H.-J.C. and M.B.P.; Funding acquisition, H.-J.C.

**Funding:** This work was supported by the research program of the Korea Institute of Energy Technology Evaluation and Planning (KETEP) funded by Ministry of Trade, Industry & Energy (20153010092090) and Incheon National University (International Cooperative) Research Grant in 2016.

**Conflicts of Interest:** The authors declare no conflict of interest.

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
