# Peer review of "Dehydration of Bioethanol to Ethylene over H-ZSM-5 Catalysts: A Scale-Up Study"

_catalysts, doi:10.3390/catal9020186_

Round 1
Reviewer 1 Report
This manuscript described an interesting study on zeolite catalysts for bioethanol dehydration. The paper is fairly well written, and results are presented in a sufficiently transparent fashion. It is particularly interesting that a link is made between lab and bench scale testing. I recommend publication in Catalysts after addressing the following points:
- The authors describe how activity and selectivity increase with increasing acid site density. In zeolite catalysis, it is also known that higher acid site densities induce a more rapid deactivation behavior. Although the conversion of ethanol seems to be quite stable under the tested conditions, the authors should include a discussion on the deactivation behavior of the tested catalysts. For this discussion, a quantification of the amount of accumulated cokes could be useful.
- Figure 2: it looks like for some temperatures, the selectivities do not sum up to 100% (e.g. at the highest temperature for 5h-1). What other products are being formed?
- Did the authors characterize the nature of the active sites in all catalysts? Do ‘active sites’ in this manuscript solely refers to Brønsted acid sites?
- The set of WHSV tested in Fig 2 and Fig 4 are different, why?
- Why is the temperature in Fig 4 260 °C instead of 250 °C.
- Page 4 line 138: “H-ZSM-5B composed with weak and medium acid sites rather 137 than strong by desilication showed a lower ethylene selectivity than H-ZSM-5A, and the selectivity to diethyl ether increased by more than two times.” à this is not the case at WHSV 2 h-1 so it needs to be mentioned for which conditions these observations are valid.
- Why wasn’t the lanthanum containing zeolite not tested in the lab scale test? Suddenly adding another catalyst modification at the end of the manuscript seems to break the flow of the paper.
Reviewer 2 Report
This article was written on the actual topic and suitable for the subject of this magazine.
But there are a number of remarks to the content of the article:
1. The article does not indicate the composition of bioethanol, were there any impurities?
2. Catalysts tested in two reactors. What is the fractional composition of the catalyst, the internal diameter of the reactor? The second reactor with an internal diameter of 10.9 mm and a height of the catalyst bed of 700 mm, but also does not indicate the fractional composition of the loaded catalyst. Does the requirement for both reactors:
Lb/dp > 50
dr/dp > 10
Lb - catalyst bed length, dP - catalyst particle diameter, dR - diameter of the reactor.
3. The temperature was measured in the reactor, was there a gradient?
4. How was ethylene selectivity calculated? This should be indicated in the article.
5. Figure 3 shows the temperature-programmed desorption profiles (TPD) of NH3, but the number of medium and strong acid sites is not indicated. How much has the amount of acid cents changed in samples?
6. It is necessary to clarify all the conditions for catalyst regeneration: the rate of temperature increase, the rate of air supply.
7. What is the main cause of catalyst deactivation? Does the texture of the catalyst change after regeneration?
8. What is the assessment of the life cycle of a catalyst?
Round 2
Reviewer 1 Report
I agree with the adaptations made to the manuscript. It can now be published.